# Pathway to a land-neutral expansion of Brazilian renewable fuel production

Luis Ramirez Camargo [1,2,3 ✉], Gabriel Castro [1,4], Katharina Gruber [1], Jessica Jewell [5,6,7], Michael Klingler [1,8], Olga Turkovska[1], Elisabeth Wetterlund[7,9] & Johannes Schmidt[1 ✉]

Biofuels are currently the only available bulk renewable fuel. They have, however, limited expansion potential due to high land requirements and associated risks for biodiversity, food security, and land conflicts. We therefore propose to increase output from ethanol refineries in a land-neutral methanol pathway: surplus $CO_2$-streams from fermentation are combined with $H_2$ from renewably powered electrolysis to synthesize methanol. We illustrate this pathway with the Brazilian sugarcane ethanol industry using a spatio-temporal model. The fuel output of existing ethanol generation facilities can be increased by 43%–49% or ~100 TWh without using additional land. This amount is sufficient to cover projected growth in Brazilian biofuel demand in 2030. We identify a trade-off between renewable energy generation technologies: wind power requires the least amount of land whereas a mix of wind and solar costs the least. In the cheapest scenario, green methanol is competitive to fossil methanol at an average carbon price of 95€ $tCO_2{}^{-1}$.

[1] Institute for Sustainable Economic Development, University of Natural Resources and Life Sciences, Vienna, Austria. [2] Electric Vehicle and Energy Research Group (EVERGI), Mobility, Logistics and Automotive Technology Research Centre (MOBI), Department of Electrical Engineering and Energy Technology, Vrije Universiteit Brussel, Brussels, Belgium. [3] Copernicus Institute of Sustainable Development, Utrecht University, Utrecht, The Netherlands. [4] Energy Planning Program, Graduate School of Engineering, Universidade Federal do Rio de Janeiro, Rio de Janeiro, Brazil. [5] Department of Space, Earth and Environment, Chalmers University of Technology, Gothenburg, Sweden. [6] Center for Climate and Energy Transformations and Department of Geography, University of Bergen, Bergen, Norway. [7] International Institute for Applied Systems Analysis (IIASA), Laxenburg, Austria. [8] Department of Geography, University of Innsbruck, Innsbruck, Austria. [9] Energy Engineering, Division of Energy Science, Luleå University of Technology, Luleå, Sweden. ✉email: l.e.ramirezcamargo@uu.nl; johannes.schmidt@boku.ac.at

Biofuels occupy an integral role in netzero emission scenarios[1], as they are the only bulk renewable alternative to liquid fossil fuels currently available[2,3]. In particular bioethanol production from sugarcane (Saccharum spp.) shows high potential for land-based climate change mitigation[4,5]. Brazil, the largest ethanol producer from sugarcane globally, has put in place a series of biofuels policies, such as the National Biofuels Policy (RenovaBio), the Agro-Ecological Zoning, and the current proposal for a credit subsidized COVID-19 Emergency Program to Support the Brazilian Sugar-Energy Sector (Pease). These measures aim to meet the growing demand for ethanol and sugar while safeguarding the renewable energy targets of Nationally Determined Contributions, as well as generating new incentives for biofuel production and technological innovation under sustainability criteria[2,6].

However, the wider impacts of such a pathway have been extensively discussed, showing the complexity of consequences of sugarcane expansion for land-use and the agricultural sector[7–9]. Sugarcane plantations expand to a large extent on pastures[10,11], where the associated land-use change can cause soil biodiversity losses[12] and structural soil degradation[13], as well as pollution and depletion of water resources[14–16]. Nevertheless, environmental concerns have been mostly linked to indirect land-use change, i.e., the process of sugarcane expansion on pastures drives the expansion of pasture areas into natural forests[17,18]. As carbon storage is higher in these areas than in pastures, indirect effects significantly limit the $CO_2$ savings potential of sugarcane ethanol[19,20] and endanger a variety of forest ecosystem services[21,22]. Furthermore, increasing competition for arable land reveals negative impacts on small-scale family agriculture and food security[23,24].

To safeguard against these negative land-use impacts of sugarcane ethanol production, several climate-related land-use policies and technological improvements have been proposed and partly implemented in Brazil[25]. The use of second-generation (2G) ethanol production technology is one technical option to increase land-use efficiency[26,27]. Under 2G ethanol production, the ligno-cellulosic part of the sugarcane, i.e., the bagasse, is used for ethanol production in addition to the sucrose. When combined with alternative hybrid sugarcane crops, which increase the amount of the fiber at the cost of the sucrose content[2], the increase in land efficiency can reach more than 50%[28]. 2G ethanol, however, comes with the down-side of less surplus electricity production compared to first-generation ethanol, since part of the bagasse which is usually combusted for electricity production is used for fermentation in the 2G process[27]. Land sparing by intensified agricultural production is another key component to increase sugarcane ethanol sustainability[29–31], targeted in particular at the restoration of degraded pastures and the integration of crop-livestock-forestry systems[15,32]. For instance, by adopting sustainable management systems, the cultivation of sugarcane on degraded pastures can have several environmental co-benefits such as improved soil health and higher soil organic carbon, as well as the maintenance or recovery of forest reserves[20,22,33]. These activities are supported by the federal government's Low Carbon Agriculture Program ABC[34], designed to reduce the overall carbon debt of sugarcane and ethanol production. While incentive-based policy programs and technologies can increase the sustainable land-use of sugarcane ethanol production, apparently the pressure on viable agricultural lands is increasing, as shown by the upward trend in deforestation rates in the Brazilian Legal Amazon[35]. Therefore, approaches to reduce the pressure on land will be beneficial.

We present a new land-neutral methanol pathway and assess by how much the fuel output can be increased if the total plantation area is fixed at current levels of ethanol production. This way, the associated negative impacts of sugarcane ethanol expansion are avoided, freeing up land for other purposes such as food and feed production or even afforestation, and therefore reducing the overall pressure on land resources. We propose to synthesize methanol using $CO_2$ from the fermentation process in ethanol plants and $H_2$ produced from electrolysis powered entirely by on-site variable renewable energy sources (VRES). The higher land efficiency of VRES in comparison to biomass increases the land efficiency of combined ethanol and methanol output significantly, while the almost clean $CO_2$ streams from ethanol fermentation allow us to produce methanol from $H_2$ and $CO_2$. This is beneficial, as methanol is much easier to handle and transport than pure $H_2$[36].

Understanding the cost and magnitudes of the required infrastructure is a key element for the implementation of the land-neutral methanol pathway. While hydrogenation of $CO_2$ to produce methanol has received significant attention in industry and academia[37], including major R&D efforts in Brazil[38], recent estimates of power-to-methanol processes show that they are significantly more costly than their fossil counterparts[39,40]. However, our proposed pathway includes important elements that can make green methanol more competitive: the size of $H_2$ and $CO_2$ storage is optimized to maximize capital utilization[41], while the almost pure and low-cost $CO_2$ stream from the ethanol production process can further reduce the cost.

Here, we illustrate the land-neutral methanol pathway in the case of Brazil, the sugarcane ethanol frontrunner. Our pathway allows for an increase in renewable fuel production without compromising the compliance with national and international environmental standards such as RenovaBio and the European Union's Renewable Energy Directive II[42]. We show that under certain plausible techno-economic assumptions, the $CO_2$ abatement cost from the land-neutral methanol pathway is competitive with other climate change mitigation technologies and that additional land-neutral fuel production potentials can reach almost 50% of current sugarcane ethanol production.

## Results

**Illustrating the land-neutral methanol pathway.** We optimize power-to-methanol processes at all Brazilian sugarcane plants in a spatially and temporally explicit way that accounts for the seasonality of sugarcane production, and the variability of PV and wind power at all plant sites (Fig. 1). $CO_2$, $H_2$, and electricity storage options are incorporated to handle the temporal asynchrony of the electricity and gas streams. The entire process is studied under a variety of technological assumptions in two scenarios. First, we differentiate between a scenario with mixed PV and wind power generation (solar–wind scenario), and a scenario with wind power generation only (wind scenario). We do not assess a PV-only scenario, as it would not improve on cost or land-use efficiency over the solar–wind scenario. In contrast, the wind scenario can improve land-use efficiency—at a higher cost—due to the lower land footprint of wind power. Second, we assess how varying techno-economic assumptions regarding annualized electrolyzer cost, efficiency of the electrolysis, and annualized cost of PV generation affect land-use, methanol cost, and system configurations (see Methods and Supplementary Table 1 for details). The production of green methanol can be easily combined with other land-efficient technologies in ethanol production, as the $CO_2$ streams from fermentation increase proportionally to the amount of ethanol produced. We therefore also explore the implications for the feasibility of the proposed pathway of ethanol production at higher land-use efficiencies.

We choose methanol as the final product, as (i) the process to derive it from $H_2$ and $CO_2$ has been intensely studied[37], (ii) it is a

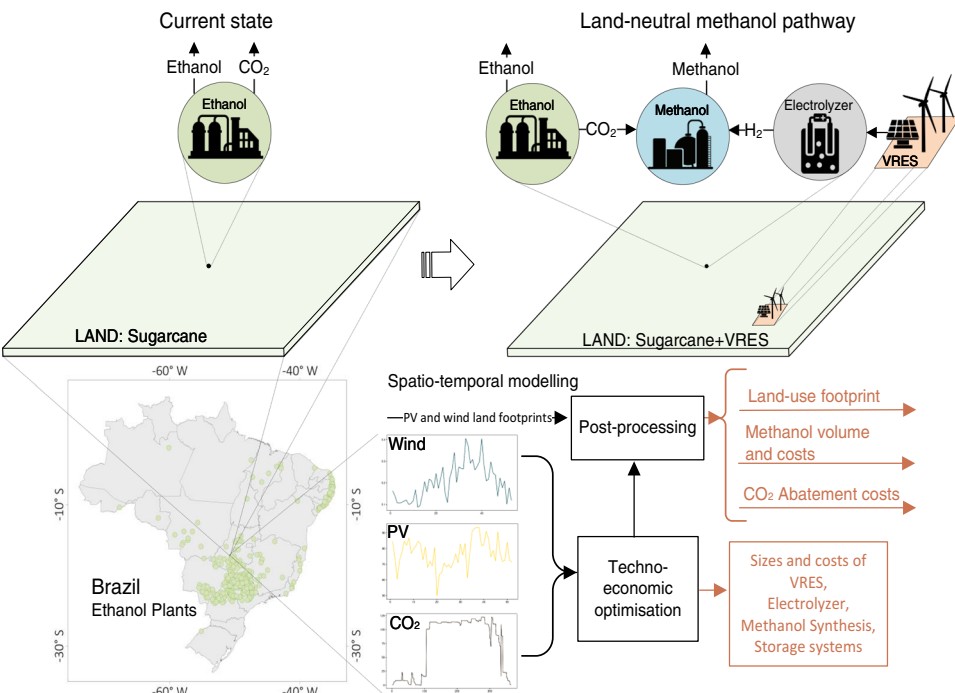

**Fig. 1 The proposed land-neutral methanol pathway.** The left top side presents the current state of fuel production at sugarcane ethanol plants, while the right top shows the new system configuration under the pathway. The bottom of the figure shows how Brazilian-specific data on the Brazilian sugarcane ethanol industry and solar and wind resource estimations are combined to assess the proposed pathway.

liquid fuel, easy to transport without additional infrastructure, and (iii) it has a variety of applications as feedstock for the chemical industry and as a transportation fuel. In particular, it can complement ethanol in blends of methanol, ethanol, and gasoline[43]. Therefore, it can be used in flex-fuel engines optimized for ethanol-gasoline blends without requiring modifications to the engine or vehicle, as well as in heavy-duty long-distance road and maritime transport[44].

**The methanol pathway strongly increases land-use efficiency.** The additional output in energetic terms increases by 43–49% when implementing the land-neutral methanol pathway. The lower value results from the solar–wind scenario with the lowest wind share, where a significant amount of PV generation capacity is installed. There is, however, high variability between the results in the solar–wind scenarios, and in the best case, the output in energetic terms increases to 47%. The upper bound of the range of 49% comes from the wind scenario and is a result of the much smaller direct footprint of wind turbines, as spacing areas between turbines can be used for sugarcane production (Methods). If, in energetic terms, the same amount of sugarcane ethanol would be produced at the average Brazilian land-use efficiency, between 23,000 and 27,000 km$^2$ of additional land would instead be required[45].

The available amount of methanol, i.e., about 100 TWh, could cover several times the domestic fuel demand for shipping[46], about 27% of the freight transport in the country[47], or around 15% of the current global methanol demand[48]. The average land-use efficiency of today's Brazilian ethanol production is 3.7 GWh km$^{-2}$. The combined land-use efficiency of methanol and ethanol production is 5.56 GWh km$^{-2}$ for the wind scenarios and 5.40 GWh km$^{-2}$ for the solar–wind scenario. While the wind scenario outperforms the solar–wind scenario in land-use efficiency, it is more expensive under all techno-economic assumptions.

The high land-use efficiency of the land-neutral methanol pathway is highlighted for all installations for two selected scenarios in Fig. 2 (see Supplementary Table 4 for details on the choice of techno-economic parameters). The figure also illustrates that land-use variability is very high for the wind scenario and lower for the solar–wind scenario. This is a consequence of highly variable wind conditions and comparably low heterogeneity in solar radiation between locations, and of the dominant impact of PV on land-use in the solar–wind scenario. In the solar–wind scenario, the share of wind is on average about 45%. Some locations have, however, much higher wind shares. These locations represent the outliers in terms of land-use efficiency for some facilities in the North-East and the South-East, where the facilities are located close to the very windy coast, allowing for a 100% share of wind power production in the solar–wind scenario. Ethanol plants, and therefore the potential for combined ethanol and methanol production, are highly concentrated in two regions: the Central-West, and the South-West, both regions with low wind resources. In these regions, the state of São Paulo is the most important area, producing 46% of total Brazilian sugarcane ethanol volume (Supplementary Fig. 1).

The relative increase in land-use efficiency—made possible by the land-neutral methanol pathway—decreases if the sugarcane-to-ethanol land-use efficiency increases (Supplementary Fig. 2: at high sugarcane-to-ethanol efficiency, substituting sugarcane by VRES will not save as much land as at a lower efficiency). The sugarcane-to-ethanol efficiency has increased by around 1% every 3 years in the period 2005–2020[49]. If efficiency gains remain at this order of magnitude, impacts on the overall efficiency of the land-neutral methanol pathway are very low. However, a technological leap such as an upgraded conversion technology (i.e., 2G ethanol), potentially combined with sugarcane breeds of higher yield[2], could allow for a one-off increase in ethanol output by 50%. At this value, the production gain implied by the land-neutral methanol pathway drops by 2 percentage points on average in the solar–wind scenarios, and by 0.2 percentage points

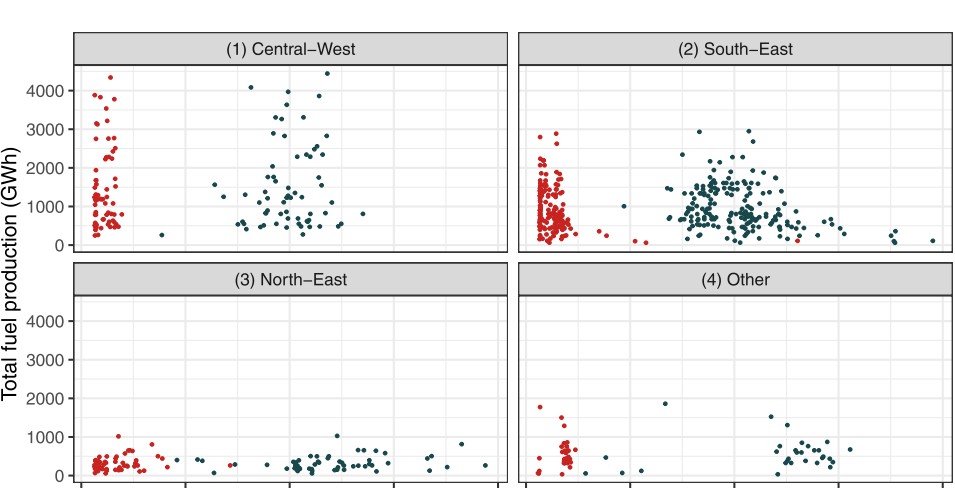

**Fig. 2 Land-use efficiency of methanol production versus total fuel production for all Brazilian ethanol facilities in (1) the Central-West, (2) the South-East, (3) the North-East, and (4) other Brazilian regions.** The panels show the land-use efficiency of methanol production measured in GWh km$^{-2}$ versus the total fuel production in that particular facility. Red points relate to scenarios using both PV and wind power, green points relate to wind power only scenarios.

in the wind scenario. We also assess how much the sugarcane-to-ethanol efficiency would need to increase to make the addition of methanol from renewable power inefficient in terms of land-use: at an increase of the ethanol land-use efficiency by about 17 times in the solar–wind scenario, and by about 137 times in the wind scenario, the total renewable fuel output if substituting sugarcane for VRES would decrease. This magnitude clearly illustrates that the land-neutral methanol pathway can provide land-use efficiency improvements under any realistic future increase in sugarcane-to-ethanol land-use efficiency.

**Methanol cost determined by electrolyzer and solar PV cost.** The methanol cost of the land-neutral methanol pathway strongly depends on uncertain costs of the technological components. We test a range of plausible cost estimates from the highest—where cost stays at today's level—to the lowest—where cost falls to around two-thirds (PV) and one-third (electrolyzers) of what they are today (Methods and Supplementary Table 1). We find that the cost of methanol in the proposed pathway is most significantly impacted by the cost of the electrolyzer, the solar PV system, and the $CO_2$ storage. On average, the cost of methanol increases by 2.8€Cent kWh$^{-1}$ when the electrolyzer cost increases from 30 to 95€ kW$^{-1}$ a$^{-1}$. The higher PV cost scenario increases the cost on average by 1.1€Cent kWh$^{-1}$, and the higher $CO_2$ storage cost scenario by 0.7€Cent kWh$^{-1}$. Varying the assumptions on the cost of electrolyzer efficiency, $H_2$ storage technologies, wind power, or the selected weather year can add up to another 14% of variation to the resulting methanol cost estimates (Supplementary Note 1). Furthermore, this is a highly capital-intensive production scheme and increases in the interest rate from 8% to 12% and 16% will increase the methanol cost by 32% and 62%, respectively (Supplementary Note 1).

The higher $CO_2$ storage cost[50–52] scenario represents above-ground storage of liquefied $CO_2$ in tanks which can be implemented at all locations. The lower $CO_2$ storage cost[53–55] represents the storage of liquefied $CO_2$ in rock caverns, which availability depends on local geological conditions that are not assessed in detail here. The spread in methanol cost across the sets of assumptions is significant but is consistently lower under

the solar–wind scenarios than the wind scenarios. The wind scenarios are about 16–43% more expensive than solar–wind combinations under the same techno-economic assumptions (Fig. 3). The wind scenario is therefore not cost-competitive and we do not discuss it in the following.

In the solar–wind scenario, the average production costs of methanol vary, due to different techno-economic assumptions, between about 7 and about 12€Cent kWh$^{-1}$, i.e., the cost increases by 70% between sets of assumptions (Supplementary Note 1). Compared to fossil methanol (4€Cent kWh$^{-1}$ [48]), our production cost estimates imply an increase in production cost between 80 and 200% (see Supplementary Note 2). As a comparison, a recent compilation of green methanol cost[48] from VRES has found an increase of 64–390% compared to a fossil methanol cost of 4€Cent kWh$^{-1}$, depending on the chosen modeling approach, techno-economic parameters, and $CO_2$ source.

Our range of carbon abatement cost estimates among all scenarios and locations is between 49 and 368€ tCO$_2$$^{-1}$, when compared to fossil production of methanol from natural gas via steam reforming and synthesis gas conversion, at a methanol price of 0.04€ kWh$^{-1}$ (see Supplementary Note 2).

The supply curves show that differences in the climate conditions for VRES generation and the load profiles of $CO_2$ have an impact on the methanol cost, i.e., when assessing variation in methanol cost within one set of techno-economic assumptions, the methanol cost of the most expensive location is 47–110% higher than that of the cheapest one, depending on which underlying techno-economic assumptions are chosen. To assess in detail the impact of climate conditions and the $CO_2$ profile on methanol cost, we run a regression model for the solar–wind scenario. We select one set of techno-economic assumptions (see Supplementary Note 3 for details) and regress the average capacity factors of PV and of wind, and the length of the $CO_2$-supply season on methanol cost. Our regression explains 82% of variation in the data. Average wind capacity factors are strongly significant and inversely related to cost: an increase in the capacity factor of 0.1 would decrease cost by 0.5€Cent kWh$^{-1}$ on average. Capacity factors of wind power vary by up to 0.63 between locations, thus wind availability can explain up to

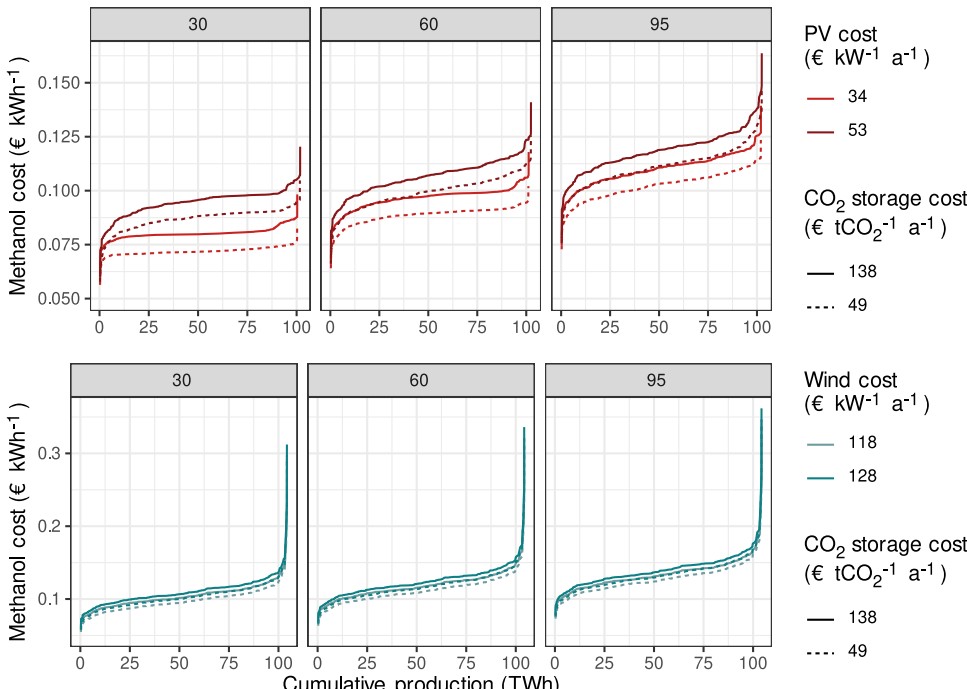

**Fig. 3 Cost–supply curves for selected scenarios and cost assumptions.** The cost assumptions are annualized. Upper: solar–wind scenario. Lower: wind scenario. From left to right: annualized electrolyzer cost assumptions (€ kW$^{-1}$ a$^{-1}$). We assume here an electrolyzer efficiency of 69%, an annualized battery storage cost of 33€ kWh$^{-1}$, and an annualized $H_2$ storage cost of 74,300€ $t^{-1}$. PV photovoltaics.

3.2€Cent kWh$^{-1}$ or 33% of variation in cost for the chosen techno-economic assumptions. Higher wind availability has two effects on cost: first, the levelized cost of electricity at those locations may be lower than for PV, thus reducing the production cost. Furthermore, wind output is also available during the night. This allows for a more continuous operation of the electrolyzers, and therefore lower electrolyzer capacities, which reduces the overall installed cost. Our regression analysis does not show a significant contribution of higher solar radiation to cost reductions. First, variation in radiation is comparably low, and second, there is a (weak) negative correlation between solar radiation and wind resources. Counter-intuitively, the longer the $CO_2$-supply period, the higher the final methanol cost. This is a consequence of comparably low $CO_2$-storage cost: if the $CO_2$ supply is concentrated in a short period of time, the $CO_2$ storage can be filled up quickly and the variation in renewable power supply can be balanced by the $CO_2$ storage at most times. However, if the $CO_2$ supply is stretched over a long period, the $CO_2$ storage needs time to fill up and, in some circumstances, renewable power supply will be higher than the currently stored $CO_2$. As storing electricity or $H_2$ is much more expensive than storing $CO_2$, higher renewable electricity generation and methanol synthesis capacities are installed at those locations to compensate for curtailed power and methanol production at the start of the season. The effect can become relatively large for extreme cases: a $CO_2$ supply season that is 100 days shorter will decrease the methanol cost by 0.8€Cent. The difference between the shortest and the longest period is 260 days, so in total, ceteris paribus, the factor could explain about 2€Cent or about 21% of differences in cost. Multi-annual $CO_2$ storage is one way of addressing this challenge, but was not assessed by us.

**PV dominates wind power, $CO_2$ storage dominates $H_2$ storage**. Here, we show the size of the different technical components used in the land-neutral methanol pathway under different techno-economic assumptions, in GW and kt, respectively (Fig. 4).

Depending on the PV and electrolyzer cost scenario, wind power capacity shows high variation. At simultaneously high PV and electrolyzer cost, the wind power capacity reaches 35 GW, close to the PV capacity in this scenario at 43 GW. Exclusively wind turbines from the International Electrotechnical Commission (IEC)-III class are built. However, at low PV and electrolyzer cost, the share of wind drops to 3 GW. Both the relative cost of electricity generation and the more favorable production profile of wind power drive this pattern. Strongly falling electrolyzer capacities under a higher penetration of wind power confirm this explanation.

There is about three orders of magnitude more $CO_2$ storage than $H_2$ storage and battery storage is not at all installed under any of the techno-economic assumptions. This follows from $CO_2$ storage being the least costly form of storage, for which reason it is used to balance the system seasonally as well as hourly. If the process is installed at all locations in Brazil, the required capacities of the components are large: PV capacities of up to 98 GW, wind power capacities of up to 35 GW, and electrolyzer capacities of between 35 and 66 GW would have to be installed, depending on the scenario. In total, $CO_2$-storage capacities of up to 10 Mt would be necessary, as a constant $CO_2$ stream has to be guaranteed throughout the year, while the fermentation process runs only seasonally. The scale of these installations is huge, e.g., for PV it is one order of magnitude larger than the currently installed PV generation capacity in the Brazilian electric system[46], and for wind power, it is up to twice the currently installed capacity. Neither $CO_2$-storage nor electrolyzer capacities are deployed at any significant scale in Brazil currently, and corresponding deployment challenges will emerge.

The mean size of PV installations at plants is 185 MW. While this is less than half the size of the currently largest PV installation in Brazil (São Gonçalo)[56], the largest installation would have a capacity of 1500 MW, which is about the scale of the largest installations globally[57]. The largest ethanol plants require storage sizes of 192 kt of $CO_2$, while the mean size is 27 kt of $CO_2$.

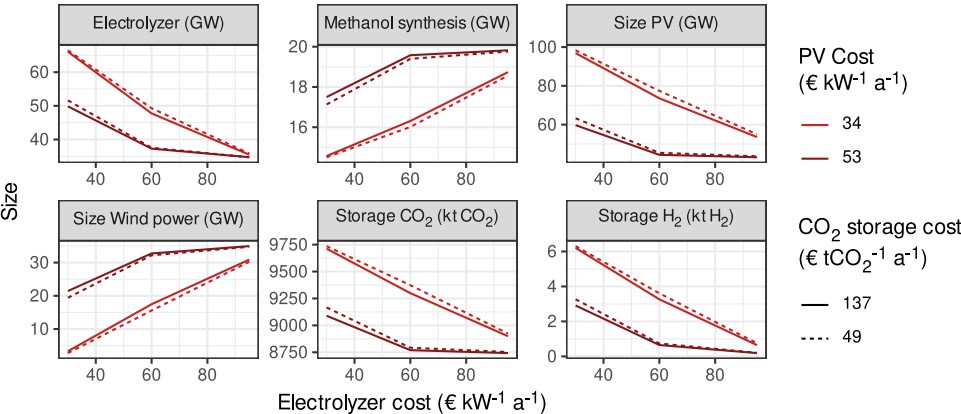

**Fig. 4 Sizes of electrolyzers, methanol synthesis, solar PV, wind power, and storage installations.** The figure shows the sum of all installed capacities at all locations in the solar–wind scenario. PV photovoltaics.

## Discussion

The implementation of the land-neutral methanol pathway would substantially decrease Brazilian $CO_2$ emissions by about 34 Mt annually, which amounts to almost 9 and 4% of total Brazilian $CO_2$ emissions related to energy conversion and land-use change, respectively[58]. Similar to biofuel production, Brazil could become a forerunner in synthetic fuel production. This could in the long-term pave the way to future fully synthetic fuel production using direct air carbon capture instead of carbon captured by biomass, thus increasing output on the same land by at least a factor of 10 compared to today's ethanol pathway[28]. An alternative approach to handle the $CO_2$ from ethanol distillation is to store it permanently underground, i.e., to implement bioenergy with carbon capture and storage (BECCS). This pathway would allow for negative emissions[59,60]. As long as fossil emissions can be substituted by green methanol, however, BECCS and the green methanol pathway have a very similar impact on atmospheric $CO_2$ levels. Furthermore, BECCS would not help alleviating the pressure on land-use generated by the sugarcane ethanol industry, and would require substantial pipeline infrastructure to move the $CO_2$ from production to permanent storage sites. Depending on local conditions and chosen parameters, recent modeling of $CO_2$ abatement cost estimates a cost range for the ethanol industry between 20 and 388€ $tCO_2^{-1}$ for BECCS[60]. In particular, for the cases where pipelines are necessary, the $CO_2$ abatement cost would be substantially higher than in the land-neutral methanol pathway proposed here.

To achieve the full potential of social, environmental, and industrial co-benefits of the land-neutral methanol pathway and to make it economically competitive, a supportive land-use policy would have to be implemented. Only under such a policy could potential rebound effects from increased demand[61] be prevented. At the same moment, such a land-neutral policy would increase biofuel prices, making our proposed land-neutral methanol pathway more competitive. Therefore, the support of Brazilian land conservation policies such as a rigorous enforcement of the Brazilian Forest Code[62] combined with additional carbon pricing is necessary for ensuring the competitiveness of our proposed pathway. Policymakers also have to consider that the highly capital-intensive pathway may lead to increasing the profit share of capital owners at the cost of the labor share, compared to ethanol production only.

There are also uncertainties for the land-neutral methanol pathway, which can only partly be addressed by today's policy and decision makers: short-term demand reductions for fuels due to the COVID-19 crisis and long-term demand reductions due to electrification of transport will have a negative impact on the

demand for liquid fossil fuels. At the same time, under strict climate change mitigation scenarios, demand for renewable fuels will increase as they are fundamental to fully decarbonize industry, heavy-duty transportation and electricity systems. The short and medium-term development of the Brazilian industry is therefore controversially discussed and a future decrease in production cannot be ruled out[63]. There is also uncertainty about the cost of future PV generation and, to an even greater extent, of electrolyzers, which strongly affect the cost-competitiveness of the land-neutral methanol pathway. The development of technological cost will partly depend on global investments in research and development and deployment of these technologies. Still, the land-neutral methanol pathway alone would increase demand for these technologies to a scale that would allow building up a national industry in the synthetic fuel sector, thus substantially lowering cost. Finally, little is known about the environmental impacts of substituting sugarcane plantations by PV and wind turbines. Presumably, most environmental impacts will decrease[64]. However, life cycle analysis shows that in some cases, the negative impacts of solar PV on water ecotoxicity exceed those of biomass-based energy[64]. Also, volant wildlife species are at risk from wind turbines[65,66], as are environmentally vulnerable areas[67]. The analysis of these uncertainties is crucial for, e.g., biodiversity conservation and can best be tackled by future research.

## Methods

**Producing methanol from surplus $CO_2$ and renewable $H_2$.** Sugarcane-to-ethanol plants in Brazil use mostly first-generation conversion technology which produces ethanol by fermentation of sugarcane juice followed by distillation. Ethanol plants are energy self-sufficient thanks to Combined Heat and Power (CHP) plants, where bagasse is burnt to produce electricity and process heat[45,68]. Bagasse can alternatively be used as feedstock for 2G biofuel production, thus increasing the fuel output. However, this comes at the cost of decreased electricity exports to the grid. We propose to upgrade existing sugarcane plants by adding a water electrolyzer, driven by VRES generation, and a methanol synthesis unit that combines $H_2$ from the electrolyzer with $CO_2$ separated from the ethanol fermentation process. We assume that the electrolyzer is not connected to the electricity grid, but that it relies on electricity generated by VRES locally. This assumption is justified as electricity demand for electrolysis would be substantial at large ethanol facilities. The transmission infrastructure available currently is not sufficient to transport such high amounts of electricity and building it would be very costly. In addition, the generation of electricity on-site is potentially less costly than paying wholesale market prices plus grid fees and could also avoid the risk of high electricity prices during years with low hydropower generation.

We develop a linear optimization model (LP) to minimize the cost of VRES-based synthetic methanol production (see supplementary Note 4). The methanol production process is an adaptation of the electrochemical methanol process model proposed by Hannula[69], and the sugarcane ethanol refinery model proposed by Gnansounou et al.[70]. The detailed model with streams of sugarcane, electricity, heat, $CO_2$, ethanol, and methanol in the facility derived from these two sources was

simplified to streams of electricity, $H_2$, and $CO_2$ as functions of the amount of produced ethanol (see Supplementary Fig. 9).

Based on the premise of energy self-sufficiency of the installation, we assume that the CHP runs always when heat is required for the ethanol production process, but that surplus electricity is not used in the electrolyzers, because assuming internal use would likely indirectly affect $CO_2$ emissions in the Brazilian electricity system due to reduced electricity exports. To make our assessments of carbon abatement independent of those indirect impacts, we do not take changes in electricity exports from the CHP or VRES into account. The model is solved over one year of operation for each installation individually, as there are no interactions between facilities. We assume that the sizing of components at the existing sugarcane refinery remains fixed, while we optimized the size of PV and wind power generation facilities, electrolyzers, and methanol synthesis reactors, as well as electricity, $H_2$, and $CO_2$ storage sizes. All processes are simulated on an hourly basis. The simulation year does not start on the 1st of January, but at the moment when the cumulative sum of the difference of instantaneous $CO_2$ stream and mean $CO_2$ stream is at its minimum: this coincides with the moment in the year when, optimally, the $CO_2$ storage is completely emptied and starts to be filled again. As production profiles for the installations are homogenous for all installations within one state, within each state the simulation year starts at the same point in time.

**A synthetic data set of the Brazilian ethanol industry**. At the moment, there is no single or consolidated source of data on ethanol generation plants in Brazil. In contrast to previous studies, we therefore developed our own consolidated database. We included data from three official sources, i.e., the Energy research company (EPE–Empresa de Pesquisa Energética), the National Agency of Oil, Natural Gas and Biofuels (ANP–Agência Nacional do Petróleo, Gás Natural e Bio-combustíveis), and the Ministry of Agriculture (MAPA–Ministério da Agricultura, Pecuária e Abastecimento). Details can be found in Supplementary Note 5.

Little is known about the temporal production profiles in ethanol plants. In all related modeling approaches (e.g. [26,45]), a certain number of hours of continuous generation is assumed, as the daily or hourly operation profile of the plants is irrelevant in these approaches. The seasonality of ethanol production, as well as intra-day operation variability, is however crucial to our approach. We therefore approximate the time profile of ethanol production using the statistics for sugarcane crushing on a state basis by the Companhia Nacional de Abastecimento (CONAB)[71]. This source includes data on the number of operation months, running days, and hours of operation per day, as well as processing volume shares by month and state. We use the volume shares to define in which months plants are operating, the total number of running days to restrict the length of the season, and accommodated the operation hours in the middle of each operating day to determine the hours of operation (e.g., a plant with 20 hours of operation runs between 02:00 and 22:00). The volume of ethanol generation per hour is equal to the total annual ethanol production divided by the number of operation days per year and the number of operating hours per day. Due to limited data availability, temporal profiles of sugarcane production only differ between states. However, we scale the level of production to the sizes of the individual plants. Assuming that sugarcane fermentation and therefore $CO_2$ formation occurs at the same pace as ethanol production, the $CO_2$ emission time series are calculated for each installation using the hourly ethanol generation profiles (see Supplementary Fig. 10).

**Photovoltaic energy generation potential and footprint**. Hourly electricity generation of PV installations is calculated for the location of each ethanol plant using PV_LIB[72] and ERA5-land[73] data. Time series are generated assuming the use of monocrystalline panels with a fixed-tilt orientation toward the north and inclination equal to the latitude. This configuration is a common approximation for installations to maximize the output of electricity per year. The approach has been validated against hourly electricity generation data of PV installations in Chile[74]. Since there are no PV installations close to ethanol plants that are available for validation, we compare the average yearly and monthly generation totals of PV electricity output calculated with our approach to SOLARGIS data[75] for each location. The correlation, relative mean bias error and relative root mean squared error are on average 0.99, 0.08, and 0.09, respectively. This implies a slight over-estimation of the PV production compared to SOLARGIS. The average capacity factors of solar PV installations at all ethanol plant locations are presented in Supplementary Fig. 11.

The land footprint of PV is estimated for seven large PV plants in Brazil, the locations of which were derived from a data set provided by ANEEL[56]. Using the measuring tool in Google Earth, the extension of these plants is measured in two different ways. In the first step, the area covered by panels is measured. In a second step, the panels and the surroundings used for roads, in-between panel space, and electrical infrastructure are also determined (detailed results are provided in Supplementary Tables 9 and 10). The number relevant for our model is the total land requirement including spacing and infrastructure. We assume this to be at about 24 m² kWp⁻¹, i.e., the average of the values from all assessed plants. This value is comparable to values in literature (see Supplementary Table 11).

**Wind energy generation potential and footprint**. Potential wind power generation at the location with the highest wind speed in a perimeter of 40 km around each ethanol plant location is simulated using a power curve-based model. As a climate data source, we use hourly ERA5 reanalysis wind speed data with a spatial resolution of 30 km and the Global Wind Atlas Version 2 with a spatial resolution of 250 m for mean bias correction at a higher spatial resolution. The model has been validated against generation data in Brazil, the United States, South Africa, and New Zealand and shows satisfactory simulation quality, i.e., the RMSE is below 0.4 for most locations (0.16 on average)[76]. On average, the modeled wind power generation underestimates observed generation by around 4% of the capacity factor in Brazil. Two turbine types in different IEC wind classes are simulated: Vestas V90 2.0 MW (IEC class II) and Siemens SWT 3.15 142 (IEC class III). Losses due to wake are not considered, assuming that the spacing between turbines is sufficiently large. Time series of average weekly capacity factors at all locations for the two turbine types are presented in Supplementary Fig. 12.

We estimate land requirements for wind turbines by measuring the area of the existing turbine pads and roads that connect them via Google Earth. The State of Rio Grande do Norte is used as a benchmark for estimating land requirements as it has the largest number of installed wind parks among all federal states in Brazil[77]. We measure four wind parks, i.e., two wind parks with a turbine capacity of 2 MW and two parks with a turbine capacity of 3 MW. Based on these parks, we estimate the average turbine pad area per MW with and without a road for the different turbine capacities. Hence, the area of sugarcane plantations that needs to be replaced by wind turbines includes the respective pad and road requirements per MW. Our estimates for 2 and 3 MW turbine capacity, 0.33 and 0.48 ha MW⁻¹ respectively, are in-line with values reported by Denholm et al.[78], i.e., an average of 0.3 ha MW⁻¹ with a standard deviation of 0.3 ha MW⁻¹ for US wind parks.

**Cost assumptions and scenarios**. The annualized cost of the individual systems is calculated assuming an interest rate of 8%, as this rate is also used by the Brazilian administration in long-term studies of the expansion of the energy systems[79,80]. In the supplementary Note 1, we also study the sensitivity of model results to the interest rate. Lifetime, CAPEX, and OPEX vary depending on the technology. The PV and wind power installation lifetime is set to 20 years. PV investment cost in the low-cost scenario is derived from Vartiainen et al.[81] and the Danish Energy Agency[82] at around 290€ kW⁻¹ for 2030. The high cost of 431€ kW⁻¹ represents the investment cost in 2020. For wind power, a similar approach is taken, using the Danish Energy Agency data[82]. Battery lifetime is set to 15 years, and CAPEX and OPEX were taken from the 2020 and 2030 scenarios in Vartiainen et al.[81].

Other technological assumptions are collected from a thorough literature review. $H_2$ storage cost is assumed based on the numbers in Papadias et al.[83] and counter-checked with other publications, which present values in a similar range[84–86]. For the estimation of $CO_2$ storage cost, we consult a range of studies[50–52,87,88]. We propose two scenarios: a low-cost scenario with underground cavern storage and a high-cost scenario with tank storage. For the calculation of cost for the model, the average of low- and high-cost scenarios supplied in two studies[50,51] together with liquefaction cost[52] are used for the tank scenario, while for the cavern scenario three studies are consulted[53–55]. The cost of capturing the $CO_2$ from the fermentation process is assumed to be 10€ $tCO_2^{-1}$ in all scenarios, which is the highest cost reported in our selection of publications (see Supplementary Table 3). We assume that alkaline electrolyzers are used. These have lower ramping capabilities than polymer electrolyte membranes. They should, however, be able to fully ramp production in less than one hour, which is compatible with ramps in renewable generation, in particular, if small buffering batteries are included in the system. Likewise, limited curtailment can help in stabilizing ramping rates. The cost for electrolyzers is highly variable in the literature and we therefore cover a wide range of cost assumptions, as outlined in Supplementary Table 1. We test the impact of different cost assumptions in 24 different scenarios presented in the main text. Additional variations, i.e., a total of 96 scenarios, are presented in Supplementary Note 1. The 24 main scenarios consist of two different VRES configuration alternatives (only wind and mixed solar and wind), three different assumptions on electrolyzer cost, two different assumptions on $CO_2$ storage cost, and two different assumptions on the cost of solar PV installations (see Supplementary Table 1). For $H_2$ and battery storage cost, we assume the upper bound on the cost range shown in Supplementary Table 1, while for wind turbines we assume the lower bound. An additional sensitivity analysis in Supplementary Note 1 also varies these cost assumptions in 96 scenarios, but the impacts were minor. Also, we test the impact of using three different weather years on results in 24 scenarios (Supplementary Fig. 3).

**Limitations and areas for future research**. The proposed spatio-temporal energy model is a perfect foresight model, as an annual supply of $CO_2$, and generation of PV and wind power are assumed to be known in advance. This makes the sizing of components fit the particular parameters perfectly. A sensitivity analysis of climate input shows that the differences in cost between different weather years are well below 10%, therefore showing that our approach is robust to climate variability.

Total $CO_2$ storage cost is significant and can increase methanol cost by up to 10%, depending on per unit storage cost assumptions. We use two cost scenarios: in the high-cost scenario, we assume the use of above-ground $CO_2$ storage tanks, which can be installed at all plant locations. The low-cost scenario however

depends on the availability of rock caverns, which depends on local geological conditions. We do not assess which plant locations do satisfy the respective geological requirements, as this needs detailed knowledge of local geology. The availability of this storage option may therefore be limited. Other storage options could reduce the amount of $CO_2$ storage. In particular, sugarcane storage could contribute to balance the strong seasonality of sugarcane availability as an alternative to $CO_2$ storage. However, the sugarcane may suffer degradation unless energy-intensive cooling is used[89].

The electrolyzers would have to be operated in an intermittent way, including quick ramping, and completely stopping operation when there is no intermittent electricity generation. We do not restrict our model to respect these technical limits, but assessed ex-post for selected runs how often electrolyzers are started and stopped. For locations that solely depend on PV, electrolyzers are stopped every night. For a 20-year lifetime, this results in 7300 starts and stops. This is in the range of starts and stops allowed by electrolyzer producers (2500–10,000)[90]. At locations that rely on solar PV only, electrolyzers that satisfy these requirements will therefore have to be chosen. The ramping implied by our model is in line with the fast ramping capabilities of alkaline electrolyzers. However, future research assessing the operation of the whole system on the level of a single plant will be required to resolve all operational questions in detail.

Moreover, we did not consider that electricity currently produced in CHP plants from the burning of bagasse could also be used to produce hydrogen, in combination with solar PV and wind power. First, the amounts of CHP electricity are minor compared to what is needed in our pathway and therefore wouldn't drastically change our results. Second, this electricity is currently used in the national grid, and using it locally instead could have indirect carbon-emission effects somewhere else in the system. We therefore opted to exclude it from our analysis.

We also neglected local water requirements for the electrolyzer, as the input was estimated to be minor[91], especially compared to the water needs of sugarcane plantations. There are also significant uncertainties associated with the approach we used to derive production profiles for ethanol plants. However, these have minor impacts on results, as significant $CO_2$ storage is deployed at all installations anyhow.

**Reporting summary**. Further information on research design is available in the Nature Research Reporting Summary linked to this article.

## Data availability

The full set of necessary input data for the optimization model and the final results presented in all figures are available under an open license on Zenodo (https://doi.org/10.5281/zenodo.6471331). Raw ERA5 and ER5-land data can be downloaded from https://cds.climate.copernicus.eu/#!/home. Similarly, the GWA data can be obtained from https://globalwindatlas.info/download/gis-files. Corresponding download scripts are also provided in the repository. Moreover, we obtained raw data from ANEEL, EPE, CONAB, ANP, and MAPA from https://www.gov.br/aneel/pt-br, https://www.epe.gov.br/pt, https://www.conab.gov.br/, https://www.gov.br/anp/pt-br/, and https://www.gov.br/pt-br/orgaos/ministerio-da-agricultura-pecuaria-e-abastecimento to compile our data set about existing ethanol sugarcane facilities. The data sets are partially geo-restricted (i.e., they require a Brazilian IP address) and we do not have the rights to re-distribute them. However, we provide a CSV file that contains the consolidated information necessary to run our code in the repository.

## Code availability

All GAMS, Python, and R code necessary to calculate the VRES potential, run the optimization model, and create figures are also available in the Zenodo repository (https://doi.org/10.5281/zenodo.6471331).

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

## Acknowledgements

We gratefully acknowledge support from the European Research Council grant "re-FUEL" ERC-2017-STG 758149 awarded to J.S. and funding the contributions of L.R.C., G.C., K.G., M.K., and O.T. G.C. was financed in part by the Coordenação de Aperfeiçoamento de Pessoal de Nível Superior – Brasil (CAPES) – Finance Code 001, E.W. by the Swedish national strategic research environment Bio4Energy, and J.J. received funding from the European Union's Horizon 2020 ERC Starting Grant programme under grant agreement

no. 950408 for Mechanisms and Actors of Feasible Energy Transitions (MANIFEST). The simulations were run on the Vienna Scientific Computing Grid (www.vsc.ac.at). We are very thankful for the support provided by the VSC team. We are grateful to Sebastian Wehrle and Marie Claire Brisbois who proofread the manuscript.

## Author contributions

L.R.C., G.C., K.G., M.K., O.T., E.W., and J.S. conceptualized the study, developed the methodology, conducted investigation, validated the results, did formal analysis, and developed the figures. L.R.C., G.C., K.G., J.J., M.K., O.T., E.W., and J.S. contributed to drafting, reviewing, and writing the manuscript. L.R.C., K.G., M.K., O.T., and J.S. did data curation. L.R.C., K.G., and J.S. wrote the software. L.R.C. and J.S. supervised the study and equally contributed as first authors to the manuscript. J.S. was also responsible for providing resources, project administration, and for funding acquisition.

## Competing interests

The authors declare no competing interests.
