## [Peer Review File · Nature Communications]

Pathway to a land-neutral expansion of Brazilian renewable fuel productionREVIEWER COMMENTS

Reviewer #1 (Remarks to the Author):

The article "A land-neutral expansion of Brazilian renewable fuel production" can be accepted as it is. The article proposes to increase output from ethanol refineries in a land-neutral methanol pathway. Surplus CO₂-streams from fermentation are combined with hydrogen from renewably powered electrolysis to synthesize methanol. The work is original and is very significant to the field. An extensive comparison to literature was provided besides a well described methodology (a detailed Supplementary Information document was provided in addition to the well organized article).

Reviewer #2 (Remarks to the Author):

This is a very interesting paper that proposes an alternative for Brazil to increase the production of biofuels without increasing deforestation. In my opinion, this is an important study that needs to be published after minor revisions.

In the line 47, the authors mention: "with a concurrent intensification of active pastures¹⁶ or through second-generation (2G) ethanol production technology^{17, 18}."

My concern is with the intensification of active pastures, these crops tend to give way only to sugar cane and this crops tend to migrate to deforestation areas, forming a new deforestation cycle (See and include: Ferrante & Fearnside - The Amazon: biofuels plan will drive deforestation. NATURE, 577: 170, 2020).

In the line 63 to 66 the authors mention: "We illustrate the land-neutral methanol pathway for the case of the sugarcane ethanol frontrunner Brazil, which has experienced rising expansion of sugarcane plantations into biodiversity hotspots of Amazon, Pantanal and Cerrado biomes during the last decade^{21, 22}".

However, Ferrante carried out a legal move that overturned the presidential decree that would release the cultivation of sugarcane in the Amazon and in the Pantanal, preventing the increase of sugarcane crops in the two biomes. See. <https://www.oeco.org.br/noticias/justica-suspende-liberacao-da-cana-no-pantanal-e-na-amazonia/>

Here I would mention that it is necessary to maintain the cultivation blockade for the Amazon and Pantanal, as the authors present an alternative that supplants this expansion over the Amazon and Pantanal. With this, the authors' result gains more strength in my view, as it presents an alternative for expanding the capacity of biofuels while helping to protect the Amazon and Pantanal biomes.

Examples of the damage caused by sugarcane on biomes that have been heavily occupied by the crop may be important here, See Ferrante et al. The matrix effect: how agricultural matrices shape forest fragment structure and amphibian composition. JOURNAL OF BIOGEOGRAPHY, 44: 1911-1922, 2017.

In line 240, the authors need to clarify what would be "land-neutrality of Brazilian biofuel production".

In line 244, the authors need to update the text, as the presidential decree that allow cultivation in Amazonia and Pantanal was revoked.

Reviewer #3 (Remarks to the Author):

The overall idea of the paper is interesting, considering the perspectives of producing methanol from CO₂/H₂ obtained from ethanol production and electrolysis process using renewable sources. However, considering this process is still in its early stages in Brazil, being developed by researchers at UNiversity of Sao Paulo (<https://www.lianerossi.org/>) (and not mentined in the paper), I suggest authors to include a deep discussion of the current situation and perspectives of developing such process.

In page 2, lines 37-43, statemens about issues related to ethanol production are quite controversial, considering that sugarcane production expansion is mainly due to increase in agricultural productivity and possible new land uses are related to degradae land, as indicated in the Sugarcane Environmental-Economic Zoning, not quoted in the paper. Please refer to previous publications indicating such aspects. Such other possibilities are mentined in the next paragraph, but the controversial statement about negative impects remains the same. It does not seems necessary to use (not realistic) negative impacts of sugarcane expansion to propse methanol production. Please refer to existing publications not mentioned in the paper (<https://www.sciencedirect.com/science/article/abs/pii/S0301421508001080>, Goldemberg, Coelho, Nastari et al. Production and Supply Logistics of Sugarcane as an Energy Feedstock. In Wang, L. (ed), "Sustainable Bioenergy Production", 2013; Coelho, S. T., Guardabassi, P. "Ethanol". In: B. D. Solomon, R. Bailis (eds.), Sustainable Development of Biofuels in Latin America and the Caribbean, DOI 10.1007/978-1-4614-9275-7_3 © Springer Science+Business Media NewYork, Coelho&Goldemberg, Sustainability and Environmental Impacts of Sugarcane Biofuels, In Sugarcane Biofuels (Tahan et al, editors), Springer, 2019)

In addition, existing perspectives for introducing BECCS (Bioenergy with carbon capture and storage) are not considered in the paper when discussing current situation, such as <https://www.prnewswire.com/news-releases/fs-innovates-with-first-beccs-bioenergy-with-carbon-capture-and-storage-project-in-south-america-301310511.html>; PERSPECTIVES FOR BECCS IMPLEMENTATION IN BRAZIL, Mascarenhas, Karen L.1,2,3; Perecin, D. 1,2,3 ; Coelho, Suani T.1,3; Meneghini, Julio R.1,3
1Research Centre for Gas Innovation, University of São Paulo, eubce 2021, Proceedings)

Based in all these considerations, I suggest the paper to have a major revision from the authors.

Reviewer #4 (Remarks to the Author):

Review of Camargo et al

The paper proposes an interesting approach to increase the land use efficiency of biofuels. It proposes a route from methanol production by capturing the almost pure stream of CO₂ from the fermentation phase of the ethanol production process and combining with H₂ produced from wind/solar-driven electrolysis. The advantage is that it leverages the low cost of solar and wind generation to produce hydrogen, avoiding the grid expansion and improvements that would be required to bring the electricity produced to market.

The paper makes a valuable contribution to maximising the resource efficiency of land while reducing the carbon intensity of the ethanol produced. The data used and the methods employed seem to be solid, and proper description of models and databases built for the research are satisfactorily documented either in the manuscript or in the SI. However, a key weakness in the argumentation for this "land-neutral" approach lies in the treatment of the underground storage step for CO₂ and H₂. The assumption of ubiquitous available underground CO₂ storage (Lines 413-417) is questionable. Although theoretical potential may exist as described in the Atlas, there is deep uncertainty about the sustained injectivity of CO₂ wells and may prove to be the limiting factor even when a technical

potential for volumetric capacity exists (Lane et al 2021).

This weakness is recognised and left for future work, but it should be considered here, even if only as a sensitivity in which locations farther than a critical distance from past or current oil and gas explorations are given very high hurdle rates that would require high rates of return to justify the initial spending. This is of particular importance because the CO₂ and H₂ underground storage is a crucial component of the viability of the approach. I explain further.

The challenge is that the injection rate needs to match the CO₂ production (capture) rate and be sustained for the life of the project. This can only be guaranteed after significant investment is made to characterise the geologic formation into which the CO₂ will be injected. That is, substantial amounts of money need to be sunk before a decision to invest in the whole project is made.

As reported by Lane et al (2021), assessments of CO₂ storage in deep saline aquifers “show that dynamic capacities modelled over multidecadal timespans were up to 20 times lower compared with capacities inferred from a static estimate assuming no constraints on injection rate. Furthermore, those dynamic studies often overlook crucial rate-limiting factors due to a lack of site-specific data. There are even real-life examples in which an injection site was identified as having a substantial and suitable storage volume, only to have the subsequent development abandoned because of injectivity limitations. In other words, knowing the available storage volume says little about how much CO₂ could be practicably sequestered in that location.”

The fact that the CO₂ volumes to be stored at each site in this land-neutral approach may be smaller can work for or against the viability of a potential project. Smaller volumes require smaller reservoirs but also means the cost of characterising the reservoir will be spread over a smaller volume of CO₂. Seeing this CO₂ storage is a mission-critical component of the project, it requires more careful treatment than simply assuming it to be ubiquitously available and ready to be used. Also, the repeated filling and emptying of the reservoir is not a feature of existing CO₂ storage projects. What happens to a project’s viability in case of a mismatch between rates that CO₂ can be injected or retrieved to/from the reservoir and the rate required to match the CO₂ production or consumption? How much CO₂ needs to be stored at each location in tons/yr? The cost estimates used (refs 58 and 59) assume CO₂ storage in the 5 MtCO₂/yr range. Is this comparable to what would be required in the proposed approach? Much smaller volumes would likely mean higher costs per ton of CO₂.

To what extent do the challenges to CCS explored in the recent paper by Lane et al (2021) constrain the potential for this land neutral approach? The key issue here is the cost of subsurface reservoir characterisation prior to project financial investment decision, which can be prohibitive and should be included in this assessment, or at least considered as a sensitivity and discussed. Regions with a history of oil and gas exploration can benefit from subsurface knowledge and this can be an advantage for plants located in those regions. The Parana basin region does not have a history of O&G production and the costs of characterisation there are likely to be higher, if not prohibitive. This would impact the majority of existing ethanol plants in Brazil, which are mostly located in the state of São Paulo.

As a critical component of the process, this CO₂ conundrum deserves as much attention as was devoted to the other components such as the electrolyser and land footprint assessments. Otherwise, it risks painting too rosy a picture of its potential.

In addition, I have a few other suggestions for improvement.

Lines 168-170: In addition to calculating carbon abatement costs by comparing to fossil methanol routes, it would also be useful to know the abatement costs per ton of the CO₂ that would otherwise be emitted from the fermentation process itself. Also, would be good to then compare to current methanol prices to get a sense of how much the value of the product can offset the costs.

Line 319: I think sugarcane is crushed, not ground. A language detail perhaps.

Line 379: Why 8% interest rate? Is this normal for power projects like the proposed? Interest rates can be quite high in Brazil, so I suggest a sensitivity analysis on this parameter to explore the range of possible interest rates in Brazil and how they affect the outcomes.

Line 392-394: I am not an expert on hydrogen production through electrolysis, but I understand from conversations with colleagues who are that although electrolysers can be ramped up in this time, repeated start/stop may present challenges. To what extent is this considered?

References:

Lane, J., Greig, C. & Garnett, A. Uncertain storage prospects create a conundrum for carbon capture and storage ambitions. *Nat. Clim. Chang.* (2021). <https://doi.org/10.1038/s41558-021-01175-7>

Response to reviewers:

We would like to thank the reviewers for taking the time to thoroughly review the manuscript. We addressed all comments and all additions are highlighted in the version of the manuscript that includes the changes. To address the main concerns of the reviewers and confirm that our results are robust and the proposed pathway viable, we have significantly changed the manuscript. In particular, we have addressed the following main issues:

- **Discussion of sustainability of sugarcane ethanol**

Reviewer #2 and Reviewer #3 have partly opposing views about the sustainability of sugarcane ethanol, and we have therefore updated our introduction to more comprehensively address the complex discussion on the sustainability of Brazilian bioethanol production. We now clearly separate between direct and indirect land-use impacts of sugarcane expansion and have expanded our discussion on policies and technologies used in Brazil to mitigate these impacts.

- **Assessment of robustness of CO₂-storage**

We made a thorough revision of the literature on CO₂ storage and reached out to experts in the field. We have to agree to the concerns of Reviewer #4 on the feasibility of intermediate CO₂ storage in saline aquifers. Consequently, we changed the assumed storage technologies used in our models. Our new results show that even when considering the storage of liquefied CO₂ in tanks, which is significantly more expensive than the originally proposed storage in saline aquifers, the final methanol cost increases by around 10% only. The advantage of this type of storage is that its deployment does not depend on local geological conditions. It can therefore be installed at all locations. This confirms that our results are robust to changing assumptions on the cost of CO₂ storage and that CO₂ storage below ground is not necessary to make our pathway viable. We have, however, in a sensitivity analysis also additionally assessed the impact of rock cavern storage on final methanol cost. The availability of this type of storage would lower final methanol cost by a little less than 10%, but it depends on the presence of particular local geological formations. At the national scale, it is impossible to assess these conditions thoroughly, and, we therefore do not differentiate locations with respect to storage availability.

- **Change in wind power technology**

Furthermore, during the process of discussing our work internally and with external experts, we were made aware that our choice of wind turbines in the initial submission was not optimal. We now included an IEC-III class wind turbine, which performs better at comparably low-speed winds. As a result, the share of wind power in some of the mixed scenarios increased substantially and has contributed to slightly reducing final methanol cost. This change, however, does not impact our conclusions.

- **New scenario runs**

Both, to accommodate the changes in wind and storage technologies, as well as to address the requirement of Reviewer #4 who proposed a sensitivity analysis on the interest rate, we had to create new scenarios and re-run all existing ones. In total, we have optimized the process in over 130,000 optimization runs, taking into account 339 locations and 384 different techno-economic parameter configurations. All figures and numbers in the manuscript were updated accordingly.

Reviewer #1:

R1-1) The article “A land-neutral expansion of Brazilian renewable fuel production” can be accepted as it is. The article proposes to increase output from ethanol refineries in a land-neutral methanol pathway. Surplus CO₂-streams from fermentation are combined with hydrogen from renewably powered electrolysis to synthesize methanol. The work is original and is very significant to the field. An extensive comparison to literature was provided besides a well described methodology (a detailed Supplementary Information document was provided in addition to the well organized article).

Thank you for the positive feedback. This is quite encouraging and reassuring for the quality of our work.

Reviewer #2:

R2-1) This is a very interesting paper that proposes an alternative for Brazil to increase the production of biofuels without increasing deforestation. In my opinion, this is an important study that needs to be published after minor revisions.

We thank the reviewer for the kind feedback.

R2-2) In the line 47, the authors mention: “with a concurrent intensification of active pastures¹⁶ or through second-generation (2G) ethanol production technology^{17, 18}.” My concern is with the intensification of active pastures, these crops tend to give way only to sugar cane and this crops tend to migrate to deforestation areas, forming a new deforestation cycle (See and include: Ferrante & Fearnside - The Amazon: biofuels plan will drive deforestation. NATURE, 577: 170, 2020).

Thanks for this comment. As pointed out in the manuscript, the discussion on the impacts of ethanol production is quite complex. This is, also confirmed by a comment by Reviewer #3. We can not fully represent the whole discussion on sugarcane ethanol sustainability in the introduction of our manuscript, as this would go beyond the scope of our publication. However, we have rewritten this part completely and have opted for a more clear separation between direct and indirect land-use change impacts of ethanol production in the introduction now:

“However, the wider impacts of such a pathway have been extensively discussed, showing the complexity of consequences of sugarcane expansion for land-use and the agricultural sector⁷⁻⁹. Sugarcane plantations expand to a large extent on pastures^{10, 11}, where the associated land-use change can cause soil biodiversity losses¹² and structural soil degradation¹³, as well as pollution and depletion of water resources¹⁴⁻¹⁶. Nevertheless, environmental concerns have been mostly linked to indirect land-use change, i.e. the process of sugarcane expansion on pastures drives the expansion of pasture areas into natural forests^{17, 18}. As carbon storage is higher on these areas than on pastures, indirect effects significantly limit

the CO₂ savings potential of sugarcane ethanol^{19,20} and endanger a variety of forest ecosystem services^{21,22}. Furthermore, increasing competition for arable land reveals negative impacts on small-scale family agriculture and food security^{23,24}.

To safeguard against these negative land-use impacts of sugarcane ethanol production, several climate-related land-use policies and technological improvements have been proposed and partly implemented in Brazil²⁵. The use of second-generation (2G) ethanol production technology is one technical option to increase land-use efficiency^{26,27}. Under 2G ethanol production, the ligno-cellulosic part of the sugarcane, i.e. the bagasse, is used for ethanol production in addition to the sucrose. When combined with alternative hybrid sugarcane crops, which increase the amount of the fibre at the cost of the sucrose content², the increase in land efficiency can reach more than 50%²⁸. 2G ethanol, however, comes with the down-side of less surplus electricity production compared to first generation ethanol, since part of the bagasse which is usually combusted for electricity production is used for fermentation in the 2G process²⁷. Land sparing by intensified agricultural production is another key component to increase sugarcane ethanol sustainability²⁹⁻³¹, targeted in particular at the restoration of degraded pastures and the integration of crop-livestock-forestry systems^{15,32}. For instance, by adopting sustainable management systems, the cultivation of sugarcane on degraded pastures can have several environmental co-benefits such as improved soil health and higher soil organic carbon, as well as the maintenance or recovery of forest reserves^{20,22,33}. These activities are supported by the the federal government's Low Carbon Agriculture Program ABC³⁴, designed to reduce the overall carbon debt of sugarcane and ethanol production. While incentive-based policy programs and technologies can increase the sustainable land-use of sugarcane ethanol production, apparently the pressure on viable agricultural lands is increasing, as shown by the upward trend in deforestation rates in the Brazilian Legal Amazon³⁵. Therefore, new approaches to reduce the pressure on land will be beneficial."

R2-3) In the line 63 to 66 the authors mention: "We illustrate the land-neutral methanol pathway for the case of the sugarcane ethanol frontrunner Brazil, which has experienced rising expansion of sugarcane plantations into biodiversity hotspots of Amazon, Pantanal and Cerrado biomes during the last decade^{21, 22}". However, Ferrante carried out a legal move that overturned the presidential decree that would release the cultivation of sugarcane in the Amazon and in the Pantanal, preventing the increase of sugarcane crops in the two biomes. See. <https://www.oeco.org.br/noticias/justica-suspende-liberacao-da-cana-no-pantanal-e-na-amazonia/> Here I would mention that it is necessary to maintain the cultivation blockade for the Amazon and Pantanal, as the authors present an alternative that supplants this expansion over the Amazon and Pantanal. With this, the authors' result gains more strength in my view, as it presents an alternative for expanding the capacity of biofuels while helping to protect the Amazon and Pantanal biomes. Examples of the damage caused by sugarcane on biomes that have been heavily occupied by the crop may be important here, See Ferrante et al. The matrix effect: how agricultural matrices shape forest fragment structure and amphibian composition. *JOURNAL OF BIOGEOGRAPHY*, 44: 1911-1922, 2017.

We thank the reviewer for this valuable update. Unfortunately, despite the suspension of the presidential decree in April 2020, the respective legal process is ongoing as of December 2021. The Federal Government appealed and therefore the future of the agro-ecological zoning (AEZ) is yet undecided. As the role of the AEZ for protecting the threatened biomes of Amazon and Pantanal is heavily discussed, as the legal future of AEZ is uncertain, and as it does not play a central role to our manuscript, we therefore opted to reduce its presence in the manuscript. We still mention it shortly in the introduction, but do not address it more extensively.

We also thank for the link to the comment and the article which refer to biodiversity impacts of sugarcane plantations. We have included them in the paragraph about sugarcane impacts in the introduction:

“However, the wider impacts of such a pathway have been extensively discussed, showing the complexity of consequences of sugarcane expansion for land-use and the agricultural sector⁷⁻⁹. Sugarcane plantations expand to a large extent on pastures^{10, 11}, where the associated land-use change can cause soil biodiversity losses¹² and structural soil degradation¹³, as well as pollution and depletion of water resources¹⁴⁻¹⁶. Nevertheless, environmental concerns have been mostly linked to indirect land-use change, i.e. the process of sugarcane expansion on pastures drives the expansion of pasture areas into natural forests^{17,18}. As carbon storage is higher on these areas than on pastures, indirect effects significantly limit the CO₂ savings potential of sugarcane ethanol^{19,20} and endanger a variety of forest ecosystem services^{21,22}.”

R2-4) In line 240, the authors need to clarify what would be "land-neutrality of Brazilian biofuel production".

Thank you for this comment. We think that the reference to "land-neutrality of Brazilian biofuel production" is misleading here and have rephrased the sentence to:

“To achieve the full potential of social, environmental and industrial co-benefits of the land-neutral methanol pathway and to make it economically competitive, a supportive land-use policy would have to be implemented.”

R2-5) In line 244, the authors need to update the text, as the presidential decree that allow cultivation in Amazonia and Pantanal was revoked.

Thank you for this comment. We have removed the respective reference to the AEZ, as it is not of core interest at that point (please see response to R2-3), and modified the text at the discussion as follows:

“Therefore, the support of Brazilian land conservation policies such as a rigorous enforcement of the Brazilian Forest Code⁶³ combined with additional carbon pricing is necessary for ensuring the competitiveness of our proposed pathway. ”

Reviewer #3:

R3-1) The overall idea of the paper is interesting, considering the perspectives of producing methanol from CO₂/H₂ obtained from ethanol production and electrolisis process using renewable sources. However, considering this process is still in its early stages in Brazil, being developed by researchers at UNiversity of Sao Paulo (<https://www.lianerossi.org/>) (and not mentined in the paper), I suggest authors to include a deep discussion of the current situation and perspectives of developing such process.

Thank you for the indication. We have had a detailed look at the provided web page and identified an article by Gothe et al. 2020 (10.1016/j.jcou.2020.101195), which describes the state of the technological development to produce methanol from CO₂ and H₂ at the mentioned University, published very recently. The process described in the publication does not focus on the source of the H₂ being renewable and the link with CO₂ from sugarcane ethanol production is narrative but not part of the

proposed analysis. Other references listed in the web page such as 10.1021/acssuschemeng.1c06008 and 10.1021/acssuschemeng.1c01678 mention the entire processes adopted by us but as part of a list of many options and only at a high level description since these documents are editorials and not full research papers. CO₂ hydrogenation into methanol is part of our proposed pathway, but not the main focus of our paper. Following Braga et al. 2020 (10.1016/j.cogsc.2020.100386, also in the list of publications of the university) “CO₂ hydrogenation into methanol has already been intensely described [36, 37, 38, 39, 40, 41, *42].” and therefore we do not see the need for an additional in depth discussion of the topic in our paper. However, we have presented economical limitations of the power-to-methanol process in the introduction and cited Gothe et al. 2020 and Braga et al. 2020 to make the readers aware of the state of the art of local developments in the field:

“While hydrogenation of CO₂ to produce methanol has received significant attention in industry and academia³⁸, including major R&D efforts in Brazil³⁹, recent estimates of power-to-methanol processes show that they are significantly more costly than their fossil counterparts^{40, 41”}

Furthermore we also modified the first statement explaining the technological choice of the proposed pathway:

“We choose methanol as the final product, as i) the process to derive it from H₂ and CO₂ has been intensely studied³⁸”

R3-2) In page 2, lines 37-43, statements about issues related to ethanol production are quite controversial, considering that sugarcane production expansion is mainly due to increase in agricultural productivity and possible new land uses are related to degraded land, as indicated in the Sugarcane Environmental-Economic Zoning, not quoted in the paper. Please refer to previous publications indicating such aspects. Such other possibilities are mentioned in the next paragraph, but the controversial statement about negative impacts remains the same. It does not seem necessary to use (not realistic) negative impacts of sugarcane expansion to propose methanol production. Please refer to existing publications not mentioned in the paper (<https://www.sciencedirect.com/science/article/abs/pii/S0301421508001080>, Goldemberg, Coelho, Nastari et al. Production and Supply Logistics of Sugarcane as an Energy Feedstock. In Wang, L. (ed), “Sustainable Bioenergy Production”, 2013; Coelho, S. T., Guardabassi, P. “Ethanol”. In: B. D. Solomon, R. Bailis (eds.), Sustainable Development of Biofuels in Latin America and the Caribbean, DOI 10.1007/978-1-4614-9275-7_3 © Springer Science+Business Media New York, Coelho&Goldemberg, Sustainability and Environmental Impacts of Sugarcane Biofuels, In Sugarcane Biofuels (Tahan et al, editors), Springer, 2019)

Thanks for this comment. As pointed out in the manuscript, the discussion on the impacts of ethanol production is complex. This is confirmed by a comment by Reviewer #2. We cannot fully represent the whole discussion on sugarcane ethanol sustainability in the introduction of our manuscript, as this would go beyond the scope of our publication. However, we have rewritten this part completely and have opted for a more clear separation between direct and indirect land-use change impacts of ethanol production in the introduction now, also emphasizing the ongoing Brazilian efforts to mitigate the negative impacts of sugarcane expansion:

“However, the wider impacts of such a pathway have been extensively discussed, showing the complexity of consequences of sugarcane expansion for land-use and the agricultural sector⁷⁻⁹. Sugarcane plantations expand to a large extent on pastures^{10, 11}, where the associated land-use change can cause

soil biodiversity losses¹² and structural soil degradation¹³, as well as pollution and depletion of water resources¹⁴⁻¹⁶. Nevertheless, environmental concerns have been mostly linked to indirect land-use change, i.e. the process of sugarcane expansion on pastures drives the expansion of pasture areas into natural forests^{17,18}. As carbon storage is higher on these areas than on pastures, indirect effects significantly limit the CO₂ savings potential of sugarcane ethanol^{19,20} and endanger a variety of forest ecosystem services^{21,22}. Furthermore, increasing competition for arable land reveals negative impacts on small-scale family agriculture and food security^{23,24}.

To safeguard against these negative land-use impacts of sugarcane ethanol production, several climate-related land-use policies and technological improvements have been proposed and partly implemented in Brazil²⁵. The use of second-generation (2G) ethanol production technology is one technical option to increase land-use efficiency^{26,27}. Under 2G ethanol production, the ligno-cellulosic part of the sugarcane, i.e. the bagasse, is used for ethanol production in addition to the sucrose. When combined with alternative hybrid sugarcane crops, which increase the amount of the fibre at the cost of the sucrose content², the increase in land efficiency can reach more than 50%²⁸. 2G ethanol, however, comes with the down-side of less surplus electricity production compared to first generation ethanol, since part of the bagasse which is usually combusted for electricity production is used for fermentation in the 2G process²⁷. Land sparing by intensified agricultural production is another key component to increase sugarcane ethanol sustainability²⁹⁻³¹, targeted in particular at the restoration of degraded pastures and the integration of crop-livestock-forestry systems^{15,32}. For instance, by adopting sustainable management systems, the cultivation of sugarcane on degraded pastures can have several environmental co-benefits such as improved soil health and higher soil organic carbon, as well as the maintenance or recovery of forest reserves^{20,22,33}. These activities are supported by the the federal government's Low Carbon Agriculture Program ABC³⁴, designed to reduce the overall carbon debt of sugarcane and ethanol production. While incentive-based policy programs and technologies can increase the sustainable land-use of sugarcane ethanol production, apparently the pressure on viable agricultural lands is increasing, as shown by the upward trend in deforestation rates in the Brazilian Legal Amazon³⁵. Therefore, new approaches to reduce the pressure on land will be beneficial"

R3-3) In addition, existing perspectives for introducing BECCS (Bioenergy with carbon capture and storage) are not considered in the paper when discussing current situation, such as <https://www.prnewswire.com/news-releases/fs-innovates-with-first-beccs-bioenergy-with-carbon-capture-and-storage-project-in-south-america-301310511.html>; PERSPECTIVES FOR BECCS IMPLEMENTATION IN BRAZIL, Mascarenhas, Karen L.1,2,3; Perecin, D. 1,2,3 ; Coelho, Suani T.1,3; Meneghini, Julio R.1,3 1Research Centre for Gas Innovation, University of São Paulo, eubce 2021, Proceedings)

Thank you for this relevant comment. We examined in detail each one of the shared references and decided to add the following statements to the discussion:

"An alternative approach to handle the CO₂ from ethanol distillation is to store it permanently under-ground, i.e. to implement bioenergy with carbon capture and storage (BECCS). This pathway would allow for negative emissions^{60,61}. As long as fossil emissions can be substituted by green methanol, however, BECCS and the green methanol pathway have a very similar impact on atmospheric CO₂ levels. Furthermore, BECCS would not help alleviating the pressure on land-use generated by the sugarcane ethanol industry, and would require substantial pipeline infrastructure to move the CO₂ from production to permanent storage sites. Depending on local conditions and chosen parameters, recent modelling of CO₂ abatement cost derives a cost range for the ethanol industry between 20 EURtCO₂⁻¹ and 388 EURtCO₂⁻¹ for BECCS⁶⁰. In particular for the cases where pipelines are necessary, the CO₂ abatement cost would be substantially higher than in the land-neutral methanol pathway proposed here"

Reviewer #4:

R4-1) The paper proposes an interesting approach to increase the land use efficiency of biofuels. It proposes a route from methanol production by capturing the almost pure stream of CO₂ from the fermentation phase of the ethanol production process and combining with H₂ produced from wind/solar-driven electrolysis. The advantage is that it leverages the low cost of solar and wind generation to produce hydrogen, avoiding the grid expansion and improvements that would be required to bring the electricity produced to market. The paper makes a valuable contribution to maximising the resource efficiency of land while reducing the carbon intensity of the ethanol produced. The data used and the methods employed seem to be solid, and proper description of models and databases built for the research are satisfactorily documented either in the manuscript or in the SI.

Thank you for the positive feedback.

R4-2) However, a key weakness in the argumentation for this “land-neutral” approach lies in the treatment of the underground storage step for CO₂ and H₂. The assumption of ubiquitous available underground CO₂ storage (Lines 413-417) is questionable. Although theoretical potential may exist as described in the Atlas, there is deep uncertainty about the sustained injectivity of CO₂ wells and may prove to be the limiting factor even when a technical potential for volumetric capacity exists (Lane et al 2021). This weakness is recognised and left for future work, but it should be considered here, even if only as a sensitivity in which locations farther than a critical distance from past or current oil and gas explorations are given very high hurdle rates that would require high rates of return to justify the initial spending. This is of particular importance because the CO₂ and H₂ underground storage is a crucial component of the viability of the approach. I explain further.

The challenge is that the injection rate needs to match the CO₂ production (capture) rate and be sustained for the life of the project. This can only be guaranteed after significant investment is made to characterise the geologic formation into which the CO₂ will be injected. That is, substantial amounts of money need to be sunk before a decision to invest in the whole project is made. As reported by Lane et al (2021), assessments of CO₂ storage in deep saline aquifers “show that dynamic capacities modelled over multidecadal timespans were up to 20 times lower compared with capacities inferred from a static estimate assuming no constraints on injection rate. Furthermore, those dynamic studies often overlook crucial rate-limiting factors due to a lack of site-specific data. There are even real-life examples in which an injection site was identified as having a substantial and suitable storage volume, only to have the subsequent development abandoned because of injectivity limitations. In other words, knowing the available storage volume says little about how much CO₂ could be practicably sequestered in that location.” The fact that the CO₂ volumes to be stored at each site in this land-neutral approach may be smaller can work for or against the viability of a potential project. Smaller volumes require smaller reservoirs but also means the cost of characterising the reservoir will be spread over a smaller volume of CO₂. Seeing this CO₂ storage is a mission-critical component of the project, it requires more careful treatment than simply assuming it to be ubiquitously available and ready to be used. Also, the repeated filling and emptying of the reservoir is not a feature of existing CO₂ storage projects. What happens to a project’s viability in case of a mismatch between rates that CO₂ can be injected or retrieved to/from the reservoir and the rate required to match the CO₂ production or consumption? How much CO₂ needs to be stored at each location in tons/yr? The cost estimates used (refs 58 and 59) assume CO₂ storage in the 5 MtCO₂/yr range. Is this comparable to what would be required in the proposed approach? Much smaller volumes would likely mean higher costs per ton of CO₂. To what extent do the challenges to CCS

explored in the recent paper by Lane et al (2021) constrain the potential for this land neutral approach? The key issue here is the cost of subsurface reservoir characterisation prior to project financial investment decision, which can be prohibitive and should be included in this assessment, or at least considered as a sensitivity and discussed. Regions with a history of oil and gas exploration can benefit from subsurface knowledge and this can be an advantage for plants located in those regions. The Parana basin region does not have a history of O&G production and the costs of characterisation there are likely to be higher, if not prohibitive. This would impact the majority of existing ethanol plants in Brazil, which are mostly located in the state of São Paulo. As a critical component of the process, this CO₂ conundrum deserves as much attention as was devoted to the other components such as the electrolyser and land footprint assessments. Otherwise, it risks painting too rosy a picture of its potential.

Thank you for this very insightful comment. We have opted to exclude CO₂ storage in deep saline aquifers completely from our assessment. After a thorough re-read of the literature and an extensive discussion with experts, we found that such storage locations will not be very practical in the context of intermediate CO₂ storage. Furthermore, storage in depleted oil or gas fields is no option for any of the ethanol plant locations, as they are mostly far away from prior oil and gas production facilities. We now instead model two different storage types, both for liquefied CO₂: above-ground tank storage and storage in rock caverns. The first can be easily implemented anywhere, no geological restrictions apply and its land-use requirements are negligible. It increases the cost of renewable methanol by ~10%, however, compared to our previous cost estimates (now reflected in the results). The second type of storage is only available in selected locations, depending on local geology. A detailed geological map of Brazil which would allow to assess suitability of locations is however not available. We therefore show the implications of having access to such - lower cost - type of storage, but cannot definitely inform about which of the locations could make use of it.

We want to stress here that although the cost of CO₂ storage was increased by a factor of almost 100 compared to the previous version, overall costs of the methanol process only increased by 10%, showing that methanol costs are relatively insensitive to CO₂ storage cost.

R4-3) Lines 168-170: In addition to calculating carbon abatement costs by comparing to fossil methanol routes, it would also be useful to know the abatement costs per ton of the CO₂ that would otherwise be emitted from the fermentation process itself. Also, would be good to then compare to current methanol prices to get a sense of how much the value of the product can offset the costs.

Please observe that the amount of CO₂ produced by (fossil) methanol combustion is equivalent to the amount of CO₂ that would be released by the ethanol facilities if the carbon was not reused. However, we now compare the mitigation cost implied by our process to the mitigation cost of carbon capture and storage from ethanol processes.

“Depending on local conditions and chosen parameters, recent modelling of CO₂ abatement cost estimates a cost range for the ethanol industry between 20€ tCO₂⁻¹ and 388€tCO₂⁻¹ for BECCS⁶⁰. In particular for the cases where pipelines are necessary, the CO₂ abatement cost would be substantially higher than in the land-neutral methanol pathway proposed here.”

Furthermore, we now more explicitly state the methanol price in our main results and included a figure that shows the relative cost of fossil methanol to the cost of green methanol in our pathway in the supplementary Figure 7.

R4-4)Line 319: I think sugarcane is crushed, not ground. A language detail perhaps.

Thanks, the word has been replaced.

R4-5) Line 379: Why 8% interest rate? Is this normal for power projects like the proposed? Interest rates can be quite high in Brazil, so I suggest a sensitivity analysis on this parameter to explore the range of possible interest rates in Brazil and how they affect the outcomes.

Thank you for this comment. We choose an interest rate of 8% as the same rate is used by EPE, the public agency responsible for long-term energy planning in Brazil. It therefore represents expected capital costs for energy related investments. However, we additionally conducted a sensitivity analysis exploring interest rates of 12% and 16%, which lead to increments in the methanol cost of 32%-62% on average. This analysis is now included in the sensitivity analysis in the supplementary material.

R4-2) Line 392-394: I am not an expert on hydrogen production through electrolysis, but I understand from conversations with colleagues who are that although electrolyzers can be ramped up in this time, repeated start/stop may present challenges. To what extent is this considered?

Thank you for this highly relevant comment. We did an ex-post analysis on starts and stops of electrolyzers and added the following paragraph to the section on limitations:

“The electrolyzers would have to be operated in an intermittent way, including quick ramping, and completely stopping operation when there is no intermittent electricity generation. We do not restrict our model to respect these technical limits, but assessed ex-post for selected runs how often electrolyzers are started and stopped. For locations which solely depend on PV, electrolyzers are stopped during every night. For a 20 year lifetime, this results in 7,300 starts and stops. This is in the range of starts and stops allowed by electrolyzer producers (2,500-10,000)⁹¹. At locations which rely on solar PV only, electrolyzers which satisfy these requirements will therefore have to be chosen. The ramping implied by our model is in line with fast ramping capabilities of alkaline electrolyzers. However, future research assessing the operation of the whole system on the level of a single plant will be required to resolve all operational questions in detail.”

REVIEWER COMMENTS

Reviewer #3 (Remarks to the Author):

The subject of the paper is important and deserves attention
Authors have developed it in a consistent way
However, I do not understand why the use of surplus electricity to produce hydrogen through electrolysis is not considered, besides solar and wind
Authors could include a paragraph discussing this possibility

Suggestion to revise the paper

Reviewer #4 (Remarks to the Author):

I thank the authors for their careful treatment of the comments provided by all reviewers, mine included. The changes introduced have made the manuscript stronger and the analysis more robust. I have no further comments and recommend the manuscript for publication as is.

Signed: Alexandre C. Köberle, Imperial College London-021-01175-7

Response to reviewers:

We would like to thank the reviewers and editor for taking the time to review the manuscript. We addressed all comments and all additions are highlighted in the version of the manuscript that includes the changes.

We thank reviewer #4 for his recommendation to publish the manuscript and addressed the only remaining comment from reviewer #3:

“The subject of the paper is important and deserves attention

Authors have developed it in a consistent way

However, I do not understand why the use of surplus electricity to produce hydrogen through electrolysis is not considered, besides solar and wind Authors could include a paragraph discussing this possibility”

The following paragraph was included in the sub-section “Limitations and areas for future work”:

Moreover, we did not consider that electricity currently produced in CHP plants from the burning of bagasse could also be used to produce hydrogen, in combination with solar PV and wind power. First, the amounts of CHP electricity are minor compared to what is needed in our pathway and therefore wouldn't drastically change our results. Second, this electricity is currently used in the national grid and using it locally instead could have indirect carbon-emission effects somewhere else in the system. We therefore opted to exclude it from our analysis